# Immunomodulation of Mesenchymal Stem Cells in Acute Lung Injury: From Preclinical Animal Models to Treatment of Severe COVID-19

**DOI:** 10.3390/ijms23158196

**Published:** 2022-07-25

**Authors:** Ju-Pi Li, Kang-Hsi Wu, Wan-Ru Chao, Yi-Ju Lee, Shun-Fa Yang, Yu-Hua Chao

**Affiliations:** 1Department of Pathology, School of Medicine, Chung Shan Medical University, Taichung 402, Taiwan; d888203@gmail.com (J.-P.L.); littleuni@hotmail.com (W.-R.C.); jasmine.lyl@gmail.com (Y.-J.L.); 2Department of Pediatrics, Chung Shan Medical University Hospital, Taichung 402, Taiwan; cshy1903@gmail.com; 3Department of Pediatrics, School of Medicine, Chung Shan Medical University, Taichung 402, Taiwan; 4Department of Pathology, Chung Shan Medical University Hospital, Taichung 402, Taiwan; 5Institute of Medicine, Chung Shan Medical University, Taichung 402, Taiwan; ysf@csmu.edu.tw; 6Department of Medical Research, Chung Shan Medical University Hospital, Taichung 402, Taiwan; 7Department of Clinical Pathology, Chung Shan Medical University Hospital, Taichung 402, Taiwan

**Keywords:** acute lung injury, acute respiratory distress syndrome, cell therapy, COVID-19, mesenchymal stem cells

## Abstract

The coronavirus disease 2019 (COVID-19) pandemic, caused by severe acute respiratory syndrome coronavirus 2 (SARS-CoV-2), has been a major public health challenge worldwide. Owing to the emergence of novel viral variants, the risks of reinfections and vaccine breakthrough infections has increased considerably despite a mass of vaccination. The formation of cytokine storm, which subsequently leads to acute respiratory distress syndrome, is the major cause of mortality in patients with COVID-19. Based on results of preclinical animal models and clinical trials of acute lung injury and acute respiratory distress syndrome, the immunomodulatory, tissue repair, and antiviral properties of MSCs highlight their potential to treat COVID-19. This review article summarizes the potential mechanisms and outcomes of MSC therapy in COVID-19, along with the pathogenesis of the SARS-CoV-2 infection. The properties of MSCs and lessons from preclinical animal models of acute lung injury are mentioned ahead. Important issues related to the use of MSCs in COVID-19 are discussed finally.

## 1. Introduction

The outbreak of the coronavirus disease 2019 (COVID-19) pandemic, with severe acute respiratory syndrome coronavirus 2 (SARS-CoV-2) as its causative pathogen, is quickly spreading and causing the collapse of healthcare systems worldwide. A vast majority of infected individuals have mild respiratory symptoms. However, approximately 15–20% of patients with COVID-19 develop severe or critically severe symptoms or complications, such as pneumonia, acute respiratory distress syndrome (ARDS), respiratory failure, sepsis and septic shock, multi-organ failure, and even death. The formation of cytokine storm is one of the hallmarks of COVID-19, and ARDS caused by cytokine storm is the major mortality factor in patients with COVID-19 [1]. In combination with traditional treatment of support care for viral diseases, a variety of strategies have been proposed to control cytokine storm in COVID-19. However, the mortality rate in critically severe patient remains high and an unmet medical need still exists for these patients.

Biologic interest in mesenchymal stem cells (MSCs), which was first described by Friedenstein et al. in 1966 [2], has risen dramatically. As is expected, clinical application of MSCs is evolving rapidly for a variety of human diseases. Along with a broad and great immunomodulatory capacity, MSCs have antiviral activities and the potential to promote tissue repair. These properties signify their potential to treat COVID-19, especially to combat against cytokine storm.

Many aspects in pathogenesis and clinical manifestations of COVID-19 have certain similarities with infectious acute lung injury (ALI). Based on currently available results of preclinical models and stage-clinical trials of ALI and ARDS, MSC-based cell therapy is proposed as a promising option in the management of COVID-19. In this review article, we summarized the potential mechanisms of action and clinical outcomes of MSC therapy in COVID-19, along with the pathogenesis of the SARS-CoV-2 infection. The properties of MSCs, especially focusing on their immunomodulatory function, and lessons from preclinical animal models of ALI are mentioned ahead. Finally, important issues related to the application of MSCs in COVID-19 are discussed. We anticipate that more patients can benefit from the therapeutic effects of MSCs in the near future.

## 2. Properties of MSCs

### 2.1. Biologic Features of MSCs

According to the consensus statement by the International Society for Cellular Therapy in 2006, human MSCs must fulfill the minimum criteria for identification based on their in vitro growth pattern, the specific surface marker expression, and the multipotent differentiation potential [3]. With fibroblast-like morphology, MSCs must be plastic-adherent when maintained in standard culture conditions. MSCs must express CD105, CD73, and CD90, and lack expression of CD45, CD34, CD14 or CD11b, CD79α or CD19, and HLA-DR. MSCs must have the capacity for trilineage mesenchymal differentiation, including osteoblasts, adipocytes and chondroblasts (Figure 1).

### 2.2. Isolation of MSCs from Various Origins

MSCs are derived from mesodermal progenitor cells. They can be isolated from a broad spectrum of tissues (Figure 1), including adult tissues (bone marrow, peripheral blood, and adipose) and fetal tissues (cord blood, umbilical cord, placenta, and amniotic fluid). MSCs were originally identified in the bone marrow [2], and thus bone marrow is considered to be their traditional source. In fact, MSCs constitute a very small percentage, about one in 3.4 × 10^4^ cells, in the bone marrow [4]. Considering efficacy and ethical issues of isolation, to find alternative cell sources with better availability is an important issue in modern cell-based therapy.

Umbilical cords are rich in MSCs which can be easily collected and cultured [5]. Most importantly, obtaining MSCs from umbilical cords is safe for both mother and baby without requiring an invasive and painful procedure. It is of interest that umbilical-cord-derived MSCs (UCMSCs) exhibit greater proliferative potential and shorter doubling time in vitro [6], indicating a shorter amount of time needed to obtain sufficient cells for clinical use. Compared to their bone marrow counterpart, UCMSCs show higher immunomodulatory potential [6], suggesting that UCMSCs are more feasible for clinical diseases with aberrant immune responses. In addition, UCMSCs express lower levels of HLA-class I molecules [7], implicating less intensity of their immunogenicity and the superiority in clinical utility. Therefore, umbilical cords represent an appealing source of MSCs and should not be discarded as medical waste.

### 2.3. Immunomodulatory Properties of MSCs: Focus on Molecular Biology

MSCs exhibit profound immunomodulatory potential, and their effects are achieved via a number of signaling pathways and a network of complicated immune responses. MSCs have the capacity to interact with immune cells in both innate and adaptive immune systems. Mechanisms of interaction are dependent on cell-to-cell contact along with the release of a variety of soluble factors. With the expression of immunosuppressive ligands such as programmed cell death protein 1 (PD-1) and Fas-ligand on their surface, MSCs can bind receptors on the surface of immune cells and result in loss of function in immune cells [8,9]. MSCs can block the differentiation of monocytes into dendritic cells and impair their antigen-presenting ability [10]. The MSC-mediated inhibitory effects may be dependent on IL-6 and hepatocyte growth factor (HGF) secreted by MSCs, and this effect can induce monocyte-derived cells to produce IL-10, which might indirectly strengthen the suppressive effects of MSCs [11]. In addition, IL-1 receptor antagonist and CCL18 secreted by MSCs may participate in promotion of macrophage polarization from M1 to M2 phenotype, involving IL-10 secretion, regulatory T cell (Treg) generation, and the immunosuppression on CD4+ T cells [12,13].

MSCs act on the adaptive immune system in various ways. Through producing soluble factors, MSCs can inhibit B cell proliferation via arrest in the Go/G1 phase of the cell cycle [14]. Furthermore, MSCs can down-regulate B cell differentiation, chemotaxis to CXL12 and CXCL13, and immunoglobulin production [14]. MSCs influence the behavior of T cells significantly, including inhibition of proliferation, cytokine secretion, and cytotoxicity. By exposure to interferon-γ (IFN-γ) and tumor necrosis factor-α (TNF-α), MSCs can secrete PD-1 ligands and lead to the suppression of T cell activation [15]. Upon interaction with dendritic cells, MSCs can regulate the balance between Th1 and Th2 cells. Through secreting transforming growth factor-β1 (TGF-β1) and indoleamine 2,3-dioxygenase, MSCs can promote the generation of Tregs, which is one of the main features of MSC-mediated immune modulation [13,16]. MSCs can induce IL-10 and prostaglandin E2 (PGE2) production and inhibit IL-17, IL-22, and IFN-γ to limit Th17 differentiation [17]. MSCs also can suppress Th17 responses by modulating the IL-25/STAT3/PD-L1 axis [18]. Under inflammatory conditions, MSCs can mediate the adhesion of Th17 cells via CCR6 and exert anti-inflammatory effects through the induction of Tregs [17].

It is interesting to note that MSCs act like a double-edged sword in regulating the immune system. There is a dual effect of MSCs on immune reactions, and nitric oxide or indoleamine 2,3-dioxygenase may act as a switch in MSC-mediated immunomodulation [19]. As a consequence, MSCs can become highly immunosuppressive upon stimulation by inflammatory cytokines but can promote immune responses under low levels of IFN-γ and TNF-α [19]. On the other hand, MSCs express high levels of Toll-like receptors (TLRs) 3 and 4 on their surface, and ligation of TLR3 and TLR4 can inhibit the ability of MSCs to suppress T cell proliferation by the downregulation of Notch signaling [20]. In the course of dangerous infections, this mechanism can attenuate the immunosuppressive activity of MSCs and restore an efficient T cell response. Therefore, MSCs have more complicated effects on the immune system than the classical role as immune suppressor cells. By sensing their surrounding environment, MSCs are receptive to local biochemical signals and orchestrate the reprogramming of immune cells to promote host defense or resolve inflammation.

### 2.4. Clinical Application of MSCs in Humans

MSCs have great therapeutic potential for numerous diseases. More than 1000 registered clinical trials on the NIH Clinical Trial Database (http://clinicaltrials.gov/, accessed on 13 July 2022) were designed to evaluate MSCs therapy for a variety of human diseases [21]. As expected, a growing body of evidence for outcomes of MSC treatment in various diseases has been reported in the literature. It is presumable that diseases with optimal efficacy can fall into two main categories. One regards immune dysfunction and/or hyperinflammatory conditions, including autoimmune diseases (systemic lupus erythematosus, rheumatoid arthritis, type 1 diabetes mellitus), inflammation diseases (Crohn’s disease, ulcerative colitis, ARDS), and graft failure or graft-versus-host disease after transplantation. The other involves tissue repair and regeneration, including degenerative diseases (osteoarthritis, degenerative disc disease, liver cirrhosis), traumatic injury (spinal cord injury, ischemic cardiomyopathy, cerebral infarction, burn wound, diabetic ulcer), congenital diseases (multiple sclerosis, osteogenesis imperfecta, muscular dystrophy), and bronchopulmonary dysplasia. The therapeutic benefits of MSCs appear to mainly involve immunomodulation and promotion of tissue repair/regeneration. In fact, MSCs may act via multiple mechanisms to achieve their therapeutic effects.

## 3. MSC Therapy in Preclinical Animal Models of ALI

### 3.1. Complexity of ALI Pathogenesis

A number of insults, infectious and noninfectious, can lead to the development of ALI. Characteristic features of ALI include loss of alveolar-capillary membrane integrity, excessive transepithelial leukocyte migration, and uncontrolled release of inflammatory mediators [22]. Following infection or trauma, upregulation in the production of proinflammatory cytokines occurs as the direct response and as a marker of ongoing pulmonary injury. Dysfunction of microvascular endothelial barriers causes an increase in capillary permeability, which permits the efflux of protein-rich fluid into pulmonary interstitium and distal airspaces. Excessive activation of neutrophils results in basement membrane destruction, increased permeability of the alveolar-capillary barriers, and release of damaging mediators to creat ulcerating lesions in adjacent tissues. Alveolar epithelial cell damage contributes to the disruption of normal fluid transport, impairment in resolution of alveolar flooding, loss of surfactant production, and disorganization of tissue repair.

It is well recognized that uncontrolled pulmonary inflammation plays a key detrimental role in ALI, and the degree of acute inflammation and alveolar epithelial injury are highly associated with the outcome of human ALI [23]. There are a number of biomarkers proposed to predict the morbidity and mortality of ALI, including those found on the epitheilum (von Willebrand factor) and endothelium (ICAM-1, surfactant protein D, and receptor for advanced glycation end-products), and those involving inflammation (IL-6 and IL-8) and coagulation (protein C and plasminogen activator inhibitor-1) [22].

### 3.2. Potential Mechanisms of MSCs in ALI: Focus on Molecular Biology

Advances in understanding the pathophysiology of ALI have led to investigations of numerous potential therapeutics. MSCs have generated a considerable interest in this filed. The application of MSCs as cell therapy for ALI is suggested to alleviate pulmonary inflammation and injury, enhance pathogen clearance, maintain endothelial and epithelial function, and augment tissue repair [24].

MSCs are demonstrated to have higher engraftment efficiencies with sites of inflammation or injury [25]. Signaling factors released by injured tissues, such as stromal cell-derived factor-1 (SDF-1), platelet-derived growth factor, HGF, and monocyte chemoattractant protein-1 (MCP-1), serve as messages for MSC migration. In response to chemokine-attractive gradients generated by injured tissues, MSCs express a number of chemokine receptors on their surface, including CCR1, CCR2, CCR4, CCR5, CCR7, CCR8, CCR9, CCR10, CXCR1, CXCR2, CXCR3, CXCR4, and CXCR5 [26]. The communications between chemoattractive cytokines and their receptors enable MSCs to home to sites of injury and provide site-specific effects. Among them, the SDF-1/CXCR4 axis is frequently emphasized as an important pathway to control MSC migration and engraftment [27,28,29,30]. In addition, adhesion molecules on MSCs can interact directly with compromised endothelial cells in the lungs and participate in the maintenance of endothelial barrier integrity by preserving endothelial barrier proteins, such as VE-cadherin, claudin-1, and occludin-1 [31].

Excessive inflammation plays a key detrimental role in the development of ALI, and MSCs have profound immunomodulatory potential. It is rational to use MSCs as a temporary solution to downregulate hyperinflammatory reactions in ALI. In animal models of ALI, MSCs have been demonstrated to decrease production of proinflammatory cytokines (IL-1α, IL-1β, IL-6, IL-8, MCP-1, MIP-2, IFN-γ, and TNF-α) and increase anti-inflammatory cytokines (IL-10) [24]. On the other hand, by which mechanisms MSCs suppress immune cells are further investigated. Corresponding to their broad immunomodulatory effects on immune cells, a variety of pathways are proposed. MSCs can reduce neutrophil migration into the alveoli as demonstrated by histopathology or MPO activity [32,33,34,35,36]. We documented that MSCs exert their immunomodulatory influence through down-regulation of MyD88-NFκB signaling [35]. MSCs are reported to induce the generation and activation of Tregs for the balance of anti and proinflammatory factors [33]. TNF-α-induced protein 6 (*TNFAIP6/TSG-6*) was highly induced in MSCs in response to lung injury and could cause a decrease in neutrophil accumulation and lung damage [37]. Increased intensity of PGE2 receptor, EP3, was found to correlate with the immunosuppressive activity of MSCs [38].

The beneficial effects of MSCs on the clearance of bacteria in ALI were demonstrated [36]. In the face of infection, MSCs can enhance phagocytosis and improve macrophage survival. Additionally, MSCs can produce substances with antibacterial activities. For example, cathelicidin LL-37 secreted by MSCs in response to bacterial challenge in ALI can cause disruption of bacterial membranes which is lethal to the microorganisms [39]. By upregulating the antibacterial protein lipocalin-2, MSCs were found to enhance bacterial clearance from the alveolar space in mice with bacterial pneumonia [40].

MSCs have great differentiation and regenerative capacities contributing to the repairment of injury tissues. A variety of factors secreted by MSCs, including keratinocyte growth factor, antiopoietin-1, HGF, and vascular endothelial growth factor, were demonstrated to improve alveolar fluid clearance, restore alveolar-capillary barrier and lung structure, and ultimately promote lung function recovery in ALI. Additionally, MSCs are able to reshape the pulmonary microenvironment by reducing the levels of profibrogenic factors, and the efficacy is associated with the correction of inappropriate endothelial/epithelial to mesenchymal transition [41].

### 3.3. Efficacy of MSC Therapy in Animals with ALI

Many studies were conducted to examine the efficacy of MSCs in animal models of ALI. ALI can be induced in experimental animals by infections, chemical injury, trauma, or pulmonary ischemia. Using lipopolysaccharides to induce ALI in rats or mice is the most common animal model, because it can mimic infectious ALI in humans and stimulate host immune responses. As for showing low immunogenic properties, using mismatched MSCs does not trigger proliferative T-cell response and rejection in the host. Therefore, MSC origins used in preclinical studies of ALI were diverse, syngeneic, allogeneic, or even xenogenic. The most common source of syngeneic MSCs was isolated from bone marrow; xenogenic MSCs were derived from human umbilical cords, adipose tissue, and bone marrow. The number of MSCs was usually around 1 × 10^6^ cells/kg per injection. However, a great diversity of treatment schedules was found in the literature.

McIntyre et al. performed a systematic meta-analysis of preclinical studies regarding MSC treatment in ALI [42]. They found that treatment with MSCs significantly decreases the overall odds of death in animals with ALI compared to untreated disease control animals (odds ratio, 0.24). MSCs were shown to reduce inflammation, enhance bacterial clearance, ameliorate organ damage, and improve survival. There were no significant short-term adverse effects observed. It is worth noting that their efficacy was maintained across different types of animal models and means of ALI induction: MSC origin, source, route of administration and preparation; and clinical relevance of the model (the number and timing of MSC administration, supportive care strategies) [42]. As suggested by preclinical models, MSC-based therapy can be a potential modality for different types of ALI and provide supportive evidence for moving toward human clinical trials.

## 4. MSC Therapy in COVID-19

### 4.1. Pathogenesis of COVID-19

The pathogenesis of COVID-19 is complex. Through an interaction between the viral spike protein and the host angiotensin-converting enzyme 2 (ACE2) receptor, viral particles of SARS-CoV-2 can enter the target cell [43]. Owing to ACE2 receptors expressing on a variety of cells throughout the body, the clinical spectrum of COVID-19 can represent multiple organ involvement and not limit to pneumonia. Additionally, a diverse repertoire of membrane proteins can act as ACE2 cofactors or alternative receptors to allow the penetration of SARS-CoV-2 into various types of cells, even those not expressing ACE2 [44].

The host’s innate and adaptive immune responses are important to control viral spread and disease progression in COVID-19. The polyfunctional factors synthesized by SARS-CoV-2 can overcome the IFN-dependent processes of innate immunity, both the production of IFN-I-III and the subsequent transmission of secondary signals for the development of antiviral cellular stress. Suppression of IFN-I-III at the outset of COVID-19 is important for viral invasion, because IFN-I-III exhibit the greatest antiviral effectiveness in this period. In the majority of patients with severe symptoms of COVID-19, the antiviral immune-related IFN response is highly impaired [45]. It is interesting to note that IFNs play an ambiguous role in the respiratory tract in the face of infection. Effective initiation of IFN production in the upper airway leads to faster viral clearance and limits viral spread. Once invading pathogens escape from the immune control, the generation of IFN increases in the lungs which may contribute to the development of cytokine storm in the latter stages of COVID-19 [46]. SARS-CoV-2 infects alveolar macrophages and enhances the production of chemoattractants for Th1. Th1 cells can release IFN-γ, which can further promote the proinflammatory activities of monocytes and macrophages. The positive feedback loop drives persistent alveolar inflammation in COVID-19 [47].

SARS-CoV-2 infection can directly affect both the innate and adaptive immune systems, leading to disordered and chaotic immune responses. The proportions of dendritic cell compartments are significantly decreased in patients with COVID-19. An increase in levels of IFN-γ and TNF-α is associated with lymphocyte apoptosis in cytokine storm syndrome. Lymphopenia, which is significantly decreased in CD8+ T cells, CD4+ Th and Tregs in the blood, is noted in severe and critical patients [48]. On the contrary, the percentage of plasma B cells and B-cell clonality are increased. In patients with severe COVID-19, the immune landscape featured a deranged IFN-α response, profound immune exhaustion with skewed T cell receptor repertoire, and broad T cell expansion [49]. Of note, excessive production of proinflammatory cytokines, referred to as cytokine storm, is a critical phenomenon for patients with COVID-19 and is associated with disease severity, T cell depletion, systemic inflammation, and widespread tissue damage.

In mild and moderate patients with COVID-19, innate and adaptive immune responses are well-coordinated, contributing to the resolution of symptoms and convalescence. In contrast, in severe cases, uncontrolled inflammatory feedback loop triggers the formation of inflammatory cytokine storm which is one of the main features in severe COVID-19 and is associated with disease progression and severity. Many studies have been conducted to determine what inflammatory mediators are elevated in patients with COVID-19, and many inflammation-associated cytokines represent as candidates for measurement, such as IL-1β, IL-2, IL-6, IL-7, IL-12, IL-18, IL-33, IFN-α, IFN-γ, TNF-α, G-CSF, IFN-γ inducible protein 10, MCP-1, macrophage inflammatory protein 1-α, and TGF-β [50,51]. Results of the cytokine profiles in different studies may be inconsistent to some degree, and the discrepancy may relate to many variables, including disease severity, baseline conditions of patients, time points of sample collection, therapeutic modalities, etc.

Pulmonary fibrosis is found as an important factor of long-term outcomes in patients with COVID-19. Acute inflammatory lesions in the lungs are generally resolved in the convalescence stage in the majority of patients with COVID-19. However, residual abnormalities on chest computed tomography scans, such as ground-glass opacity and strip-like fibrosis, can been found in about 40% of patients at 3 months after recovery and 25% at one year [52,53]. Fibrosis-related genes, including *ACE2*, *TGFB1*, *CTGF*, and *FN1*, were found to increase transcription after the SARS-CoV-2 infection and induce lung fibrosis [54]. The process of endothelial/epithelial to mesenchymal transition may play a prominent role in the development of post-COVID-19 fibrosis [55]. Older patients with severe COVID-19 or ARDS might fare worse after the initial recovery from COVID-19 and have a higher risk of the persistent sequela of lung fibrosis.

### 4.2. Potential Mechanisms of MSC Therapy in COVID-19

The respiratory tract is usually the primary target organ of the SARS-CoV-2 infection, and many aspects in pathogenesis and clinical manifestations of COVID-19 have certain similarities with infectious ALI and ARDS. Autopsies of patients who died of critical COVID-19 showed the pathological characteristics of pulmonary lesions, including diffuse alveolar damage, interstitial inflammatory cell infiltration, protein-rich exudates, hyaline membrane formation, endothelial cell membrane destruction, mural fibrosis, and microcystic honeycombing in the lungs [56,57,58]. These findings provide the rationality of MSC treatment for COVID-19 based on the results of preclinical models of ALI and clinical trials of ARDS [59], although the specific molecular mechanisms underlying therapeutic effects of MSCs require further research. Many factors involve disease progression in COVID-19, including direct viral toxicity, cell death, endothelial dysfunction, dysregulation of immune responses, and tissue fibrosis. As expected, the immunomodulatory, tissue repair, and antiviral and anti-fibrotic properties of MSCs signify their potential in the treatment of COVID-19.

ARDS caused by cytokine storm is the major cause of death in patients with COVID-19 [1]. Therefore, inhibition of cytokine storm may be the key in the management of patients with COVID-19. It is important to note that the formation of cytokine storm provides a hyperinflammatory microenvironment to trigger immunomodulation of MSCs. In turn, the powerful immunomodulatory capacity from MSCs may reduce or even eliminate the cytokine storm in COVID-19, and consequently improve outcomes. Nevertheless, the changes in inflammatory index profiles following MSC therapy were inconsistent, as previously described. On the other hand, the lung is the most predominant organ for MSC homing. Following administration, MSCs are primarily trapped in the capillary beds of the lungs. Although, MSCs are able to migrate to diverse sites during the development of the multi-organ injury and dysfunction accompanied by the existence of COVID-19 pneumonia. Indeed, MSCs are not only useful in treating COVID-19 pneumonia, but also have beneficial impacts on the extrapulmonary complications. Figure 2 summarizes the pathogenesis of COVID-19 and potential mechanisms of MSC therapy in COVID-19.

### 4.3. Clinical Efficacy of MSC Therapy in COVID-19

The safety and efficacy of transplanted MSCs for COVID-19 were evaluated in many clinical trials. As of June 2022, there were 95 clinical trials registered on the NIH Clinical Trial Database (http://clinicaltrials.gov/, accessed on 13 July 2022) when searching for “MSCs and COVID-19“. Registered trials of MSC therapy in COVID-19 with completed or recruiting status were summarized in Table 1 [21]. Currently, the majority of these trials were in phase I, I/II, or II. Around half of the trials were completed, but only a few have reported their results. Therefore, knowledge about this issue also has to be derived from reports on treating a small number of patients.

Nevertheless, current preliminary clinical results of MSC-based therapy show some favorable outcomes for severe and critically severe patients with COVID-19. A meta-analysis by Kirkham et al. demonstrated that the risk of death at the study endpoint is lower in patients receiving MSC therapy (risk ratio, 0.18), although follow-up differed [60]. Promising results regarding MSC therapy in COVID-19 included: (1) patients were well-tolerated without significant and severe adverse effects; (2) improvements were observed in clinical symptoms and pulmonary functions, such as oxygenation and PaO_2_/FiO_2_ ratio; (3) downregulating cytokine storm by MSCs can be noted by decreasing circulating levels of proinflammatory cytokines and laboratory parameters (C-reactive protein, procalcitonin, ferritin, and D-dimer), increasing lymphocyte count, and resolving inflammatory lung lesions; (4) MSC-treated patients had a faster recovery time, a shorter hospital stay, and a lower mortality rate. According to the main goals of ongoing studies and trial reports, MSCs seem promising as an effective treatment option to alleviate the effects of COVID-19 [61].

## 5. Issues Related to Clinical Application of MSCs in COVID-19

### 5.1. Potential Risks of MSC Therapy

Despite a growing body of evidence suggesting the safety of MSC therapy, MSC administration is not completely free of risks. Adverse events include infusion reactions, allergic reactions, secondary infections, viral reactivation, and thromboembolic events.

A number of questions need to be answered regarding optimal manufacturing and the quality of MSCs for clinical use. In the management of acute indications such as COVID-19, cryobanking may remain a prerequisite. Because the number of MSCs obtained from the donor is usually insufficient for use and MSCs have a great propensity for in vitro expansion, passaged cells are used extensively in clinical practice. In vitro expansion of MSCs may be associated with genetic instability and changes in cell behavior. This highlights the importance of guaranteeing conditions for MSC culture, using cells without extensive population doublings, and monitoring the karyotyping of MSCs. On the other hand, the use of cryopreserved MSC products, which commonly contain dimethyl sulfoxide, may cause acute toxicity syndrome, such as skin reactions, headache, dizziness, nausea, vomiting, and allergic reactions. Premedication with antihistamines can prevent or alleviate these symptoms.

The risk of secondary infection is potentiated by the use of fresh products. The risk of herpesvirus transmission by transplantation of MSCs from healthy seropositive donors is low, because no viral DNA was detected in MSCs. However, MSCs may be susceptible to infection if infused into patients with viremia based on the in vitro finding that cytopathological effects and intracellular viral antigens can be found in MSCs after infection of cytomegalovirus and herpes simplex virus type 1 [62]. In addition, there is also a theoretical possibility that a state of immunosuppression induced by MSCs would allow pathogens to evade the immune response, especially in critically ill patients.

### 5.2. Optimal Treatment Protocols

Many issues related to the application of MSCs should be further explored, and consensus and guidelines are required to maximize their therapeutic efficacy. For example, the choice of a time window for MSC administration can directly affect treatment efficacy. Early initiation of MSC treatment is associated with a higher extubation rate in patients receiving mechanical ventilation. Furthermore, MSC transfusion for severe patients prior to intubation was found to reduce the risk of disease progression and mortality [41].

In the clinical trials registered in the treatment of COVID-19, MSCs derived from umbilical cords were used most frequently, followed by adipose tissue (Table 1). In our experience, UCMSCs could promote hematopoietic engraftment after hematopoietic stem cell transplantation [63,64] and treat refractory graft-versus-host disease, bronchopulmonary dysplasia, and COVID-19 effectively and safely [65,66]. Cryopreserved UCMSC products from MSC banks have great advantages over fresh products to treat COVID-19 in terms of their feasibility for clinical application and off-the-shelf availability, because they are fully characterized and confirmed sterile before infusion. Typically, these products are transported and thawed at the bedside, and it is convenient for clinical utility.

Intravenous administration has been the most utilized route of administration. Following intravenous administration, a majority of delivered MSCs rapidly reach the lungs and lodge in the pumonary vascular bed. MSCs may enter directly to reach a higher concentration in the lungs via intratracheal administration. However, its use retrained by the poor feasibility of administering fluids to hypoxemic patients. Of most importance, the risk of environmental contamination with aerosols should be seriously considered in the management of patients with infectious diseases, such as COVID-19. There is little controversy that administration of MSCs intravenously is feasible to treat COVID-19 and is much safer for the environment and care providers.

As shown in Table 1, MSCs were administrated a single time or in multiples, and the doses ranged from 0.5 × 10^6^ to 4 × 10^8^ cells/kg per injection. A dose of 1 × 10^6^ cells/kg per injection was most commonly used. In vitro, we found that the immunosuppressive effects increased with the increasing cell dose of MSCs in a dose-dependent manner [65]. It has yet to be determined whether the therapeutic effects also increase when infusing more MSCs. Additionally, the risk of thromboembolism may be increased if too many cells are infused once a time. A higher dose of MSCs may be needed to treat critically ill patients with COVID-19, but high doses greater than 12 × 10^6^ cells/kg should be used with caution [67]. Tracking studies using labeled MSCs demonstrate that most MSCs are cleared within 24–48 h. MSCs appear to act as a hit-and-run attack, and therefore multiple injections may be needed to achieve efficient effects. The injection interval of 3 days may be feasible for the acute phase in COVID-19.

## 6. Conclusions

MSC therapy brings hope to combating COVID-19 during a global pandemic. The general mechanisms of action include immunomodulatory function, tissue repair enhancement, and antiviral activities. Current preliminary results have shown some favorable outcomes for severe and critically severe patients with COVID-19. However, a variety of challenges related to its use still lay ahead. Further studies are warranted.

## Figures and Tables

**Figure 1 ijms-23-08196-f001:**
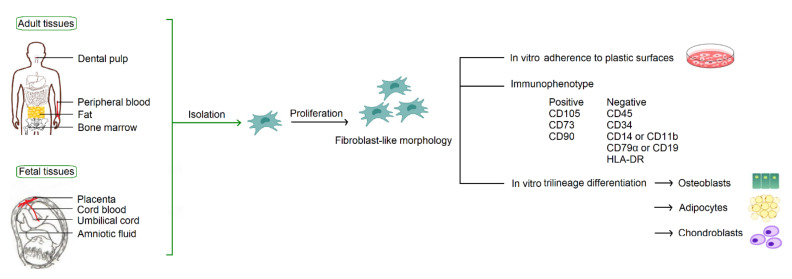
Common sources of MSCs and the minimum criteria for identification by the International Society for Cellular Therapy.

**Figure 2 ijms-23-08196-f002:**
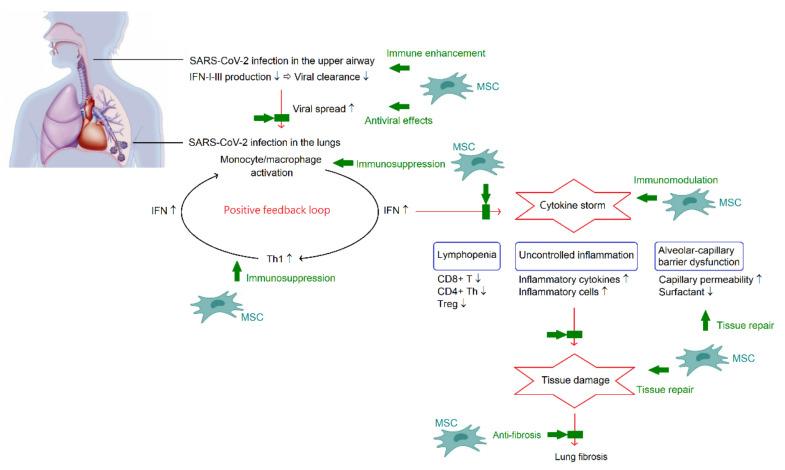
A schematic overview of the pathogenesis of COVID-19 and potential mechanisms of MSC therapy in COVID-19. In the acute phase, SARS-CoV-2 can synthesize polyfunctional factors and overcome the IFN-dependent processes in the upper airway. Once escaping from the immune control, SARS-CoV-2 can infect alveolar macrophages and enhance chemoattractant production for Th1. Th1 cells can release IFN-γ leading to further activation of monocytes and macrophages. The positive feedback loop plays an important role in the development of cytokine storm, which is a critical phenomenon of COVID-19 and is associated with lymphopenia, acute lung injury, and systemic inflammation. Multiple functions of MSCs involves in combating COVID-19. There is a dual effect of MSCs on regulating the immune system, enhancement of immune reactions to promote viral clearance and immunosuppression to alleviate cytokine storm. Along with direct antiviral activities and regenerative potential of tissue repair and anti-fibrosis, MSCs appear to be beneficial in COVID-19.

**Table 1 ijms-23-08196-t001:** Clinical trials of MSC-based therapies in COVID-19 registered on the NIH Clinical Trial Database.

Registration ID	Title	Phase	MSC Administration	MSC Source	Enrollment	Location	Status
NCT04713878	Mesenchymal stem cells therapy in patients with COVID-19 pneumonia	-	1 × 10^6^/kg, iv, on days 0, 2, 4	Unknown	21	Turkey	Completed
NCT04898088	A proof-of-concept study for the DNA repair driven by the mesenchymal stem cells in critical COVID-19 patients	-	iv, three times with 30 days intervals	Unknown	30	Turkey	Completed
NCT04611256	Mesenchymal stem cells in patients diagnosed with COVID-19	Phase 1	1 × 10^6^/kg, iv, once	AT	20	Mexico	Recruiting
NCT04336254	Safety and efficacy study of allogeneic human dental pulp mesenchymal stem cells to treat severe COVID-19 patients	Phase 1/2	2 × 10^7^/kg, iv, on days 1, 4, 7	Dental pulp	20	China	Recruiting
NCT04288102	Treatment with human umbilical cord-derived mesenchymal stem cells for severe corona virus disease 2019 (COVID-19)	Phase 2	4 × 10^7^/kg, iv, on days 0, 3, 6	UC	100	China	Completed
NCT04349631	A clinical trial to determine the safety and efficacy of Hope Biosciences autologous mesenchymal stem cell therapy (HB-adMSCs) to provide protection against COVID-19	Phase 2	iv, 5 times	HB-adMSCs (AT)	56	USA	Completed
NCT04905836	Study of allogeneic adipose-derived mesenchymal stem cells for treatment of COVID-19 acute respiratory distress	Phase 2	1 × 10^6^/kg, iv, on days 0, 2, 4	COVI-MSC (AT)	60	USA	Recruiting
NCT04348435	A randomized, double-blind, placebo-controlled clinical trial to determine the safety and efficacy of Hope Biosciences allogeneic mesenchymal stem cell therapy (HB-adMSCs) to provide protection against COVID-19	Phase 2	5 × 10^7^/kg, iv, on weeks 0, 2, 6, 10, 14	HB-adMSCs (AT)	55	USA	Completed
NCT04625738	Efficacy of infusions of MSC from Wharton jelly in the SARS-CoV-2 (COVID-19) related acute respiratory distress syndrome	Phase 2	1 × 10^6^/kg on days 0 and 0.5 × 10^6^/kg iv, on days 3, 5	UC	30	France	Completed
NCT05132972	Allogenic UCMSCs as adjuvant therapy for severe COVID-19 patients	Phase 2/3	1 × 10^6^/kg, iv, on days 0, 3, 6	UC	42	Indonesia	Recruiting
NCT04573270	Mesenchymal stem cells for the treatment of COVID-19	Phase 1	Single dose, iv	UC	40	USA	Completed
NCT04382547	Treatment of COVID-19 associated pneumonia with allogenic pooled olfactory mucosa-derived mesenchymal stem cells	Phase 1/2	NP	Olfactory mucosa	32	Belarus	Completed
NCT04355728	Use to UC-MSCs for COVID-19 patients	Phase 1/2	1 × 10^8^/kg iv, on days 1, 3	UC	24	USA	Completed
NCT04888949	A study of ADR-001 in patients with severe pneumonia caused by SARS-CoV-2 infection (COVID-19)	Phase 2	1 × 10^8^/kg, iv, once a week, total 4 times	AT	20	Japan	Recruiting
NCT04362189	Efficacy and safety study of allogeneic HB-adMSCs for the treatment of COVID-19	Phase 2	1 × 10^8^/kg, iv, on days 0, 3, 7, 10	HB-adMSC (AT)	53	USA	Terminated
NCT04535856	Therapeutic study to evaluate the safety and efficacy of DW-MSC in COVID-19 patients	Phase 1	1 × 10^7^/kg or 1 × 10^8^/kg, iv, once	DW-MSC	9	Indonesia	Completed
NCT04565665	Cord blood-derived mesenchymal stem cells for the treatment of COVID-19 related acute respiratory distress syndrome	Phase 1	iv	Cord blood	70	USA	Recruiting
NCT04903327	Study of intravenous COVI-MSC for treatment of COVID-19-induced acute respiratory distress	Phase 2	1 × 10^6^/kg, iv, on days 0, 2, 4	COVI-MSC (AT)	100	Brazil	Recruiting
NCT05433298	Mesenchymal stem cells for the treatment of patients with COVID-19	Phase 1/2	1 × 10^6^/kg, iv, once	UC	60	Brazil	Recruiting
NCT04366323	Clinical trial to assess the safety and efficacy of intravenous administration of allogeneic adult mesenchymal stem cells of expanded adipose tissue in patients with severe pneumonia due to COVID-19	Phase 1/2	8 × 10^6^/kg, iv, two doses	AT	26	Spain	Completed
NCT04629105	Regenerative medicine for COVID and flue-elicited ARDS using Lomecel-B	Phase 1	1 × 10^8^/kg, iv, three doses	Lomecel-B	70	USA	Recruiting
NCT04494386	Umbilical cord lining stem cells (ULSC) in patients with COVID-19	Phase 1/2	1 × 10^8^/kg, iv, once	UC	60	USA	Recruiting
NCT04780685	A phase II study in patients with moderate to severe ARDS due to COVID-19	Phase 2	NP	BM	40	USA	Recruiting
NCT04345601	Mesenchymal stromal cells for the treatment of SARS-CoV-2 induced acute respiratory failure (COVID-19 Disease)	Phase 1/2	1 × 10^8^/kg, iv, once	BM	30	USA	Recruiting
NCT04522986	An exploratory study of ADR-001 in patients with severe pneumonia caused by SARS-CoV-2 Infection (COVID-19)	Phase 1	1 × 10^8^/kg, iv, once a week, total 4 times	AT	6	Japan	Completed
NCT05126563	Randomized double-blind phase 2 study of allogeneic HB-adMSCs for the treatment of chronic post-COVID-19 syndrome	Phase 2	iv, on weeks 0, 2, 6, 10	HB-adMSCs (AT)	80	USA	Recruiting
NCT04392778	Clinical use of stem cells for the treatment of COVID-19	Phase 1/2	1 × 10^6^/kg, iv, on days 0, 3, 6	UC	30	Turkey	Completed
NCT04361942	Treatment of severe COVID-19 pneumonia with allogeneic mesenchymal stromal cells (COVID_MSV)	Phase 2	1 × 10^6^/kg, iv, once	BM	24	Spain	Completed
NCT03042143	Repair of acute respiratory distress syndrome by stromal cell administration (REALIST)	Phase 1/2	4 × 10^8^/kg, iv, once	UC	75	United Kingdom	Recruiting
NCT04400032	Cellular immuno-therapy for COVID-19 acute respiratory distress syndrome	Phase 1/	2.5 × 10^7^/kg or 5 × 10^7^/kg or 9 × 10^7^/kg, iv, three times	BM	9	Canada	Completed
NCT04333368	Cell therapy using umbilical cord-derived mesenchymal stromal cells in SARS-CoV-2 related ARDS	Phase 1/2	1 × 10^6^/kg, iv, on days 1, 3, 5	UC	47	France	Completed
NCT05286255	Mesenchymal stromal cells for COVID-19 and viral pneumonias	Phase 1	iv, on days 1, 3	UC	10	USA	Recruiting

AT: adipose tissue; BM: bone marrow; iv: intravenous injection; MSC: mesenchymal stem cell; NP: not provided; UC: umbilical cord.

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
