# Peer review of "Immunomodulation of Mesenchymal Stem Cells in Acute Lung Injury: From Preclinical Animal Models to Treatment of Severe COVID-19"

_ijms, 2022, doi:10.3390/ijms23158196_

Round 1

Reviewer 1 Report

The authors have considerably improved the article.

1. Angiopoietin-1 spelling should be corrected in section 3.2.

2. Please add references for the first paragraph of section 3.3.

Reviewer 2 Report

The authors responded to the reviewers' comments and made the suggested changes.

This manuscript is a resubmission of an earlier submission. The following is a list of the peer review reports and author responses from that submission.

Round 1

Reviewer 1 Report

This article by Pi Li et al reviewed the immunomodulatory effects of mesenchymal stem cells in acute lung injury and in severe COVID-19 patients. This topic is interesting as the research community is in quest of potential therapeutic options for covid-19 patients who are affected severely. MSC therapy is one of the most engrossing (also debatable) therapeutic options for diseases primarily driven by inflammation/immune cells. Here in this review, the authors have covered the properties of MSCs in general and touched on sources of MSCs used in clinical trials and in pre-clinical animal models. They have also reviewed various immunomodulatory effects of MSCs at the molecular level, especially in acute lung injury models. However, considering the topic and availability of research on the same, the authors have reviewed the topic very broadly. Also, recent reviews on the same topic (especially MSCs in COVID-19 as therapeutics) have covered many important details and the mechanisms (PMID: 35103115, PMID: 35279630, and PMID: 34497264). Authors have less focused on the main topic (in fact a very broadly explaining paragraph was written on MSCs in COVID-19). Similarly, MSCs in ALI was poorly discussed with missing key details such as the source of MSCs, dosage, type of pre-clinical model used and adverse effects found, if any.

Reviewer 2 Report

The review work presented by Ju-Pi Li and co-workers titled “Immunomodulation of Mesenchymal Stem Cells in Acute Lung Injury: From Preclinical Animal Models to Treatment of Severe COVID-19” is well written, clear, and easy to read. The topic is very interesting and therefore, it adds clustered information to the subject area of infectious diseases mediated by coronavirus with a focus on the SARS-CoV2 infection. It is a cutting-edge area and we still do not have drugs against the COVID-19 disease, therefore any other therapeutic approach is very important to contrast future coronaviruses to avoid pandemic.  In particular, the author performed a very well-conceived overview of the antiviral activities and the immunomodulatory potential of mesenchymal stem cells in contrasting COVID-19 diseases.

Minor,

Define in the abstract all the abbreviations.

I suggest adding an MSCs figure/picture/scheme in section 4, subsection 4.2 for having a graphical reading of the topic.